# Ecotoxicology Evaluation of a Fenton—Type Process Catalyzed with Lamellar Structures Impregnated with Fe or Cu for the Removal of Amoxicillin and Glyphosate

**DOI:** 10.3390/ijerph20247172

**Published:** 2023-12-13

**Authors:** Lorena Lugo, Camilo Venegas, Elizabeth Guarin Trujillo, Maria Alejandra Diaz Granados-Ramírez, Alison Martin, Fidson-Juarismy Vesga, Alejandro Pérez-Flórez, Crispín Celis

**Affiliations:** 1Department of Chemistry, Research Line in Environmental and Materials Technology (ITAM), Pontificia Universidad Javeriana, Carrera 7 No. 43–82, Bogotá 110231, Colombia; d.moralesl@javeriana.edu.co (L.L.); alison.martin@javeriana.edu.co (A.M.); alejandroperez@javeriana.edu.co (A.P.-F.); 2Department of Microbiology, School of Sciences, Microbiological Quality of Water and Sludge (CMAL), Pontificia Universidad Javeriana, Carrera 7 No. 43-82, Bogotá 110231, Colombia; c.venegas@javeriana.edu.co (C.V.); guarin.e@javeriana.edu.co (E.G.T.); diazgranadosr.mria@javeriana.edu.co (M.A.D.G.-R.); vesga.f@javeriana.edu.co (F.-J.V.)

**Keywords:** bioassays, delaminated clays, double-layer hydroxides, Fenton-type processes, inhibition, mutagenicity, toxicity

## Abstract

Antibiotics and pesticides, as well as various emerging contaminants that are present in surface waters, raise significant environmental concerns. Advanced oxidation processes, which are employed to eliminate these substances, have demonstrated remarkable effectiveness. However, during the degradation process, by-products that are not completely mineralized are generated, posing a substantial risk to aquatic ecosystem organisms; therefore, it is crucial to assess effluent ecotoxicity following treatment. This study aimed to assess the toxicity of effluents produced during the removal of amoxicillin and glyphosate with a Fenton-type process using a laminar structure catalyzed with iron (Fe) and copper (Cu). The evaluation included the use of *Daphnia magna*, *Selenastrum capricornutum*, and *Lactuca sativa*, and mutagenicity testing was performed using strains TA98 and TA100 of *Salmonella typhimurium*. Both treated and untreated effluents exhibited inhibitory effects on root growth in *L. sativa*, even at low concentrations ranging from 1% to 10% *v*/*v*. Similarly, negative impacts on the growth of algal cells of *S. capricornutum* were observed at concentrations as low as 0.025% *v*/*v*, particularly in cases involving amoxicillin–copper (Cu) and glyphosate with copper (Cu) and iron (Fe). Notably, in the case of *D. magna*, mortality was noticeable even at concentrations of 10% *v*/*v*. Additionally, the treatment of amoxicillin with double-layer hydroxides of Fe and Cu resulted in mutagenicity (IM ≥ 2.0), highlighting the necessity to treat the effluent further from the advanced oxidation process to reduce ecological risks.

## 1. Introduction

Advanced oxidation processes (AOPs) provide a promising solution for eliminating recalcitrant compounds, such as emerging contaminants [1]. These compounds are identified by their existence in extremely low concentrations and resistance to conventional physico-chemical or biological treatment methods used by various wastewater treatment plants. These substances are currently not regulated by any existing legislation and can disrupt the balance of aquatic ecosystems, causing harmful effects on human health [2].

Emerging pollutants, including pharmaceuticals and pesticides, have received particular attention due to the significant increase in their use worldwide in recent years [3,4]. Antibiotics, for example, have been heavily researched for their removal from various environmental matrices, such as soil, sludge, and water, due to the identified correlation between their presence and the proliferation of antibiotic-resistant genes. In 2021, antibiotic-resistant bacteria caused more than 700,000 deaths [5]. Herbicides can have harmful impacts on non-target organisms and humans, including skin cancer and the development of neurodegenerative diseases [6,7].

Photocatalytic and ozonation processes are among the most researched advanced oxidation processes for their effectiveness in removing various recalcitrant compounds [8]. Nevertheless, ecotoxicity studies have designated the effluents generated from these processes as being harmful to some organisms, in addition to being costly processes [9,10]. Therefore, alternative methods that were previously examined [9,11] for traditional contaminant elimination are now under investigation. Fenton-like processes are a standout alternative, with proven high efficiency when utilizing heterogeneous catalysts like transition metals supported on lamellar structures [11]. These processes mitigate the drawbacks of conventional Fenton processes, including a limited pH range and challenges with catalyst and aqueous solution separation, while also being cost-effective and avoiding the use of environmentally harmful metals such as gold and platinum during extraction [12].

Bioassays are used to evaluate potential environmental impacts and are an important tool for assessing the toxicity of chemical compounds present in the environment that may have adverse effects on various organisms. Through bioassays, it is possible to evaluate the ecological impact of these compounds on the environment. A wide range of organisms—animals (invertebrates and vertebrates), bacteria, microorganisms, and algae—are used to perform the bioassays. Among these, the most commonly used species as bioindicators to assess ecotoxic effects include *Lactuca sativa* as a plant model [13,14,15], *Daphnia magna* as an animal indicator [15,16], *Vibrio fisheri* as a bacterial model [17,18], and *Selenastrum capricornutum* as a microalgae [17,19]. In addition, modified bacteria that can respond to chemicals present in the environment are used to determine mutagenic effects at the molecular level. The Ames test is used to determine whether a compound is mutagenic [20,21].

Despite the advances in AOPs and their potential to remove recalcitrant compounds from wastewater, there have been few studies to evaluate the toxicity of effluents resulting from these treatments to various organisms. Existing studies have primarily focused on percentage removal, transformation, and degradation pathways, making it difficult to understand the ecological impact of these effluents. In recent years, there has been a growing awareness of the importance of evaluating the ecotoxicity of these effluents since they are directly discharged into surface waters, and their by-products may cause imbalances by affecting different organisms. Therefore, the present study aims to evaluate the toxicity generated by treating water containing amoxicillin or glyphosate with a Fenton-type process catalyzed by lamellar structures impregnated with Fe or Cu, using bioassays on *Daphnia magna*, *Lactuca sativa*, *Selenastrum capricornutum*, and mutagenicity using the Ames test.

## 2. Materials and Methods

### 2.1. Preparation of the Catalysts

Four catalysts were evaluated in the Fenton process for removing amoxicillin and glyphosate pollutants. Delaminated clays and double-layer hydroxides supported the Fe and Cu phases. Bentonite clay was selected and modified to be used as a support (Bentonitas de Colombia S.A.). First, hominization was carried out with a NaCl solution for 24 h with constant agitation. The conductivity was then adjusted to 100 S/cm, and polyvinyl alcohol (PVA) was added. The mixture was refluxed for 24 h and was then subjected to microwave treatment (20 min, 70 °C, 30 bar) and calcination (400 °C) [22].

In the case of the double-layer hydroxides, a coprecipitation synthesis method was used, in which the solutions of 0.1 M Mg (NO_3_) and 0.1 M Al (NO_3_) were added to an aqueous mixture of Na_2_CO_3_ (2.5 times in excess with respect to the moles of Mg^2+^) with constant stirring (200 rpm for one hour). The mixture was then subjected to microwave irradiation (20 min, 180 °C, 30 bar) with an aging time of 24 h. The mixture was then washed with distilled water until the conductivity reached 100 µS/cm. This was followed by drying at 90 °C for 24 h and finally calcination at 500 °C for 6 h [23]. For the synthesis of two catalysts, the supports were wet impregnated with Fe (III) and Cu (II) metal nitrates (each metal separately) with constant stirring (220 rpm for 24 h) to maintain a concentration of 5% of the active phases. Thus, catalysts supported on the delaminated clay (DC-Fe and DC-Cu) and those supported on double-layer hydroxides (DLH-Fe and DLH-Cu) were obtained.

### 2.2. Fenton-Type Process Water Samples

The water samples studied were the effluents remaining after the degradation of the pollutants using a Fenton-type process catalyzed by the solids we discussed in Section 2.1. The samples were obtained using a 250 mL solution containing the contaminant commercial amoxicillin (Aldriston, Bogotá, Colombia, 500 mg; 10 mg/L) and the active ingredient Roundup glyphosate (Bayer S. A., Leverkusen, Germany, 5 mg/L) was mixed with 0.125 g of catalyst in a Bach-type glass reactor with constant stirring (200 rpm), a constant airflow of 2 mL/min, hydrogen peroxide of 2 mL/h (0.1 M), temperature of 20 °C, and atmospheric pressure. The pH was adjusted to 7, and the reaction time was 2 h. At the end of the reaction, 150 mL was collected and filtered to be later diluted at 75, 50, 25, 10, and 1% *v*/*v* to perform the bioassays with *Lactuca sativa* (*L. sativa*), *Daphnia magna* (*D. magna*), and the Ames test. In addition, 0.5% and 0.025 *v*/*v* dilutions were made for the *Selenastrum capricornutum* (*S. capricornutum*) studies because it has been reported to have greater sensitivity compared to other models [24,25]. Based on previous results in this study, it was observed that there was still inhibition at 1%. The concentration of the contaminant was determined by high-performance liquid chromatography coupled to triple quadrupole mass spectrometry (MS-TQ) with electrospray ionization (ESI) on a Shimadzu Nexera X2—LCMSTQ 8060 instrument (Columbia, MD, USA).

### 2.3. Acute Toxicity Bioassays

#### 2.3.1. Seed Germination—*Lactuca sativa*

*L. sativa* variety Great Lake Batavia was used as a plant indicator according to the method described by Dutka [26]. Briefly, 25 seeds of similar size, shape, and color were placed per plate on a #3 filter paper (Whatman, Maidstone, UK) impregnated with 4 mL of sample in a Petri dish and incubated at 22 ± 2 °C in the dark for 5 days. Each dilution was performed in duplicate. After the incubation period, the length of the root zone was measured in millimeters (mm) to determine the inhibition effect. The results were expressed as percentage inhibition, where the lowest concentration (%*v*/*v*) affected the root growth. A concentration of 20 mg/L Zn^2+^ was used as a positive control, and reconstituted hard water was used as a negative control [17,27].

#### 2.3.2. Algal Growth—*Selenastrum capricornutum*

The test was developed according to the EPA method [28], as modified by Cifuentes [29]. Each of the six concentrations (%*v*/*v*) and the positive and negative controls were performed in triplicate. Each of the dilutions and the control were inoculated with the strain *S. capricornutum* N°1648 (UTEX, Houston, TX, USA, also known as *Pseudokirchneriella subcapitata*, at an initial cell density of 104 cells/mL. The cultures were then incubated at 23 ± 2 °C with a light intensity of 4300 lux ± 10 and continuous shaking at 100 ± 10 rpm for 96 h. Chromium (Cr^6+^) was used as a positive control at a concentration of 0.3 mg/L, and reconstituted hard water as a negative control. The results were expressed as percentage inhibition where the lowest concentration (%*v*/*v*) affected the organism [17,27].

#### 2.3.3. Acute Toxicity Bioassay with *Daphnia magna*

The animal toxicity test with *Daphnia magna* (*D. magna*) was as described by Dutka (1989) [30]. The test was performed in 30 mL plastic containers with 25 mL volume of each of the dilutions. Ten 24-h-old neonates were placed in each dilution, and each dilution was run in triplicate. After 48 h of incubation at 21 (±1) °C with a photoperiod of 16 h light/8 h dark and a light intensity of 800 (±100) lux, the number of dead organisms was recorded. Chromium (Cr^6+^) at a concentration of 0.15 mg/L was used as a positive control, and reconstituted hard water was used as a negative control. Results were expressed as percent lethality, where the lowest concentration (%*v*/*v*) affected the organism [17,27].

#### 2.3.4. Determination of the Mutagenicity Index (MI) Using the Ames Test

The Ames test used genetically modified strains of *Salmonella typhimurium* to detect mutagenic compounds that cause base substitution changes in DNA [31]. Strain TA98 (hisD3052, uvrB, rfa, pKM101, Moltox^®^, Boone, NC, USA) has a spontaneous reversion rate of 30–50 revertants/plate, and strain TA100 (hisG46, uvrB, rfa, pKM101, Moltox^®^) has a spontaneous reversion rate of 120–200 revertants/plate. The two *Salmonella* strains were grown in nutrient broth No. 2 (Oxoid, Basingstoke, UK) for 24 h at 37 ± 2 °C, 140 ± 10 rpm, to a concentration of 10^9^ cells/mL.

For testing, 0.1 mL of each concentration was mixed with 0.1 mL of strain TA98 or TA100 and 3.5 mL of histidine-biotin (0.5 mM) supplemented Top Agar. The resulting solution was homogenized and poured onto a Petri dish containing Minimal Agar. All assays were performed in triplicate. The mutagenic activity was determined after 120 h of incubation and exposure of the strains to the sample dilutions at 37 ± 2 °C, and reported in CFU/mL as revertant.

Mutagenic index (MI) values greater than or equal to two (≥2.0) are considered mutagenic [32]. Sodium azide (0.0375 mg/L) and 2-nitrofluorene (7.5 mg/L) were used as positive controls for strains TA100 and TA98. Broth No. 2 (Oxoid, Basingstoke, UK) was used as a negative control.

### 2.4. Data Analysis

Statistical analysis was conducted using IBM SPSS Statistics version 26. To determine whether there was a correlation between the evaluated treatments and the response results in the bioassays, a one-way analysis of variance (ANOVA) was performed, with a significance level of *p* < 0.05. The supports for the variance analysis (ANOVA) are provided in Appendix A.

## 3. Results and Discussion

### 3.1. Effluent Characterization

Due to the potential effects of amoxicillin and glyphosate on various organisms and the complex interplay of physicochemical factors, it was necessary to characterize the effluent after treatment with catalysts supported on delaminated clay (DC-Fe; DC-Cu) and double-layer hydroxides (DLH-Fe; DLH-Cu). It should be noted that not all processes resulted in the complete removal of the contaminants. Table 1 shows the removal percentages of amoxicillin and glyphosate contaminants and shows that some treatments removed 100% of the compounds. Therefore, solutions of each of the contaminants in all bioassays (Amx 10 mg/L and Gly 5 mg/L) were evaluated without any treatment, along with the blanks of each bioassay, to determine if the compounds produced toxicity prior to treatment. The amounts of Fe and Cu leached during the Fenton-type process can also be observed since, depending on the support of the catalysts and the reactions, leaching of the active phases can be generated, resulting in the presence of iron and copper in solution, which may affect different organisms depending on their concentrations. Finally, the table shows the final pH of each of the reactions that were later evaluated in the different bioassays. The table also shows the percentages of inhibition of *L. sativa* and *S. capricornutum*, the lethality of *D. magna*, and the mutagenicity index, which will be analyzed in the following sections and corresponds to the undiluted samples.

### 3.2. Acute Toxicity Bioassays

#### Seed Germination—*Lactuca sativa*

To evaluate the potential phytotoxic effects, the bioassay was performed with *L. sativa*. Figure 1 shows the germination inhibition percentages of treated and untreated wastewater containing amoxicillin and glyphosate. For both cases of treatment with glyphosate and amoxicillin, a significant statistical difference was observed in the response obtained by *L. sativa*. As shown in Figure 1A, the untreated effluent, that is, the 10 mg/L amoxicillin solution, produced inhibition from the most concentrated to the most diluted dilution, with higher percentages compared to the effluents treated with different catalysts. The results indicated that both the untreated and the treated effluents produced inhibition of germination; this is because amoxicillin by itself can induce oxidative stress in plants, generating different phytotoxic effects [33]. Furthermore, in the case of the treated effluents, the toxicity could be attributed to the by-products, which is consistent with the observations of Busto [34], where both amoxicillin itself and the by-products (amoxicillin penicilloic acid, amoxicillin diketopiprazine, amoxilloic acid, and 3-(4-Hydroxyphenyl)-2(1H)-pyrazinone) showed toxicity in plant organisms [34].

The results show that all processes were phytotoxic, one of the possible reasons being the presence of different reactive oxygen species (ROS) generated during the Fenton-type reaction and responsible for the oxidation of the contaminants [35]. As reported by Caverzan [36], the presence of singlet oxygen (1O_2_), superoxide (O_2_), hydrogen peroxide (H_2_O_2_), and hydroxyl radicals (OH) can cause cell damage due to oxidative stress that can be developed by protein modification and aldehyde generation [36,37]. This is consistent with that reported by Minshra [4], where amoxicillin together with various ROS can generate chemical stress leading to low protein synthesis [4].

Regarding the results of the phytotoxicity assays, it can be observed that the processes catalyzed with Fe-based catalysts showed the highest inhibition percentages at different concentrations, while those catalyzed with double-layer hydroxides impregnated with Cu showed less than 10% inhibition, and at a concentration of 75% *v*/*v*, no inhibition of germination was observed. However, this behavior is completely opposite to that shown in Figure 1B, which corresponds to the results of the samples containing glyphosate. The effluents treated with the different catalysts showed higher percentages of germination inhibition than those generated by the untreated effluent (5 mg/L glyphosate solution); this behavior was observed for most of the samples, except for the samples that had been catalyzed with LDH-Cu, which at a concentration of 25% *v*/*v* showed a lower percentage of inhibition than the sample that had not been treated. It can also be observed that the samples catalyzed with double-layer hydroxides at low concentrations (1% *v*/*v*) no longer showed phytotoxicity. This effect may be due to the residual glyphosate that could not be removed by the Fenton-type process catalyzed by the Fe and Cu double-layer hydroxides since, under the initial conditions, removal of more than 70% of the glyphosate was not achieved. According to the literature, glyphosate is highly efficient at both low and high concentrations because regardless of the concentration, it can inhibit the shikimic acid pathway and consequently inhibit the production of tryptophan, promoting a deficiency of auxins, which are growth regulators [38].

Another possible cause of germination inhibition is related to the compounds added in the formulations of commercial pesticides that allow and enhance the activity of the herbicide, and about which there is no knowledge of what happens to them during the process [39]. In the study carried out by Utzig [40], who evaluated the ecotoxicity of the oxidation of a pesticide, they showed that the by-products generated by the oxidation were responsible for the phytotoxicity in the seeds of *L. Sativa*, and in the study by Skanes (2021), they showed that the possible by-products generated apart from glyphosate are aminomethylphosphonic acid (AMPA) and phosphate [40,41].

In other studies on water treatment, *L. sativa* exhibited a toxicity rate of 12% at concentrations of 75% (*v*/*v*) after undergoing a photocatalysis process with TiO_2_ [42]. Furthermore, in studies conducted with galvanic systems, galvanic Fenton, and hydrogen peroxide treatments, the water showed toxicity for *L. sativa* in both treated and untreated waters (EC50, 29.57%) [43].

### 3.3. Algal Growth—Selenastrum capricornutum

In this study, the phytotoxic effects of different effluents of the treatments were evaluated using the *S. capricornutum* algal model. As shown in Figure 2A, the 10 mg/L amoxicillin solution resulted in inhibition percentages ranging from 13% to 5% for effluent concentrations of 100% and 75% (*v*/*v*), respectively. Previous studies have reported that antibiotics can have negative effects on algae, either due to the presence of the parent compound or the formation of toxic metabolites [44,45]. According to Fu [46], amoxicillin has a negative effect on the growth of *P. subcapitata* algae, unlike other antibiotics that also have an effect on the synthesis of the wall and do not produce this effect.

In contrast, the effluent with Gly 5 mg/L showed a total inhibition up to a concentration of 50% (*v*/*v*), after which its inhibitory effect decreased until it showed a 5% effect at a concentration of 0.025% (*v*/*v*) (Figure 2B). This observed negative effect of glyphosate at 5 mg/L is consistent with the results reported by Fernández [47] in relation to the concentration of glyphosate used, 4.7 mg/L, in which a negative incidence was still present both in the growth, in the production of biomass, and in the structures of *S. capricornutum*. The toxicity of glyphosate formulations to aquatic and terrestrial organisms has been reported to be moderately to slightly toxic [48], and its degradation to intermediates increases toxicity levels [49,50]. In addition, remediation processes have been reported to reduce the toxicity generated by antibiotics, resulting in lower levels of toxicity, according to tests with the microalga *C. vulgaris* [51]. However, at low concentrations of toxicity, algae could grow using some elements present in the commercial formulation of the contaminant [51]. *S. capricornutum,* compared to the glyphosate treatment, presented only statistical differences for this compound; however, at high concentrations (>10% *v*/*v*), the inhibition response (100%) was similar between them.

For the compounds LDH-Cu Amx, DC-Cu Amx, and LDH-Fe-Gly, an overall effect was observed up to a concentration of 10% *v*/*v*. The removal percentages for each of the treatments were 100%, 80%, and 30% (Table 1). It should be noted that the leachate concentration achieved for DC-Cu-Amx is 8.2 mg/L, which is higher than the concentrations of LDH-Cu-Amx (0.06 mg/L) and LDH-Fe-Gly (0.6 mg/L). Based on the above results, it can be assumed that treatments with high leachate levels, such as DC-Cu-Gly and DC-Cu-Amx, and low removal efficiencies, such as LDH-Fe-Gly and DC-Fe-Gly, would have different toxicity results than treatments with low leachate levels and removal efficiencies close to 100%. Studies have shown that ecotoxicity levels increase with factors such as temperature, contaminant concentration, oxidants, catalysts, and the presence of by-products from partial mineralization [35]. These factors may contribute to the observed inhibitory effects on *S. capricornutum*. For LDH-Cu-Gly, the degree of inhibition did not change when the concentration was changed. The concentration of leachate found was 0.3 mg/L, and 71% of the glyphosate was removed. This finding may explain the observed inhibition levels in *S. capricornutum*.

Despite the fact that a decrease in toxicity is obtained in all effluents with DC and LDH between concentrations of 1% to 0.025% *v*/*v*, an inhibitory response of 40% to 3% on the microalgae was still observed (Figure 2). Regarding the inhibitory effect that *S. capricor-nutum* presented against the LDH-Fe effluents of both amoxicillin and glyphosate, it was observed that despite being at low concentrations (1% to 0.025% *v*/*v*), effects were still present negatively on the vegetal model (Figure 2). The inhibition effect is similar to that obtained by Koba-Ucun [52], where they observed an effect or reduction in growth at low concentrations (100 mg/L) through the catalytic process with Zn-Fe LDHs. On the other hand, the use of other catalysts and similar compounds (Cu-Mg-Fe LDHs) at low concentrations (10 mg/L) also showed inhibition [53]. The possible causes of this effect are due to the fact that the compounds interacting with the organisms cause the release of metal ions from the (NM) nanomaterials, generation of oxidative stress, adsorption, absorption, and disruption of the barriers or walls of the microalgae [54,55,56].

It is important to mention that the only two effluents that no longer showed inhibition at concentrations of 0.5% and 0.025% (*v*/*v*) correspond to LDH-FeAmx, DC-Cu-Amx, and DC-Fe-Gly (Figure 2). The degree of affectation observed for both glyphosate and amoxicillin, along with the LHD treatment on *S. capricornutum* is because this compound has a suppressive effect on photosynthetic activity, cellular chlorophyll production, physical interactions, and oxidative stress that occurs between the algae and the compound [53]. The effect or inhibitory process is proportional to the level of LDH concentrations; however, it has been found that at 10 mg/L and 1.5 mg/L, there are inhibitory effects on the algae *S. quadricauda* [53].

The difference between *L. sativa* and *S. capricornutum* (Figure 1 and Figure 2) evaluated in this study shows a difference in inhibition behavior, mainly between concentrations from 100% to 25%, since the microalgae still showed total inhibition for this type of solution. According to some studies, *S. capricornutum* is one of the organisms within the group of algae that show greater sensitivity to toxic agents, making it one of the best models within its group [24,25]. Differences or similarities in sensitivity to toxicants may also exist between organisms within the same phylogenetic group. For example, *S. capricornutum* shows greater sensitivity to some classes of herbicides [57] compared to other compounds evaluated within the algal group [58]. On the other hand, it has also been shown that there is no major difference in the use of *S. capricornutum* and *L. sativa* when compared to different sources of surface water and wastewater treatment plant (WWTP) effluent [13].

### 3.4. Acute Toxicity Bioassay with Daphnia magna

*D. magna* showed no mortality rates for the amoxicillin 10 mg/L solution after 48 h. It has been reported that *D. magna* is relatively resistant to the effects of antibiotics such as amoxicillin, which is why relatively low acute toxicity is induced for this compound [35]; however, for the DC-Fe-Amx, LDH-Fe-Amx, and LDH-Cu-Amx effluents, 100% mortality was observed up to a concentration of 75% (*v*/*v*). Therefore, the mortality rates decreased from a concentration of 50% (*v*/*v*) to a mortality of 0% at a concentration of 1% (*v*/*v*) (Figure 3A). In the case of the DC-Cu-Amx effluent, a mortality rate of 100% occurred up to a concentration of 50% *v*/*v* (Figure 3A). These results are similar to those reported by Koba-Ucun [52], where acute toxicity is observed with the LDH lamellar structure impregnated with Zn-Fe catalysts, even at low concentrations of the compound (50–200 mg/L) for a period of 48 h.

These results suggest that the presented toxicity is due to the percentages of incomplete removal of amoxicillin; in the case of DC effluents with Fe and Cu catalysts, the removal percentages were 80 and 89%, respectively. Several studies have reported the formation of potentially hazardous by-products during the degradation of amoxicillin by advanced oxidation processes (AOPs) such as the Fenton process [35]. These by-products may be more harmful and toxic to the environment and public health than the original contaminant. Therefore, the negative effect observed in the DC effluents, in contrast to the null toxicity observed with amoxicillin at 10 mg/L, may be attributed to the formation of these by-products.

However, for the LDH effluents impregnated with Fe and Cu, a complete mineralization of 100% was observed, which means that the LDH treatment was more effective compared to DC in removing the contaminant. However, the levels of toxicity were similar compared to the DC effluent, so the generalized negative effect can be explained by the oxidative stress that the effluent induces in *D. magna* and, as a consequence, the production levels of intracellular carbohydrates and proteins are reduced due to the inability of the organism to maintain the metabolism of biomolecules such as pigments, proteins, lipids, carbohydrates and DNA [59]. Similarly, amoxicillin has been reported to have neurodegenerative capacities in aquatic macroinvertebrates similar to *D. magna*, such as *Brachionus calyciflorus* and *Corbicula flumibea* [60].

In addition, for the 5 mg/L Gly solution and the LDH-Cu-Gly effluent, a negative effect with 100% mortality was observed only at a concentration of 100% *v*/*v*. Therefore, the mortality rates decreased from 75% (*v*/*v*) concentration to 0% mortality at 1% (*v*/*v*) concentration (Figure 3B). For the DC-Fe-Gly effluent, 100% mortality was observed up to 75% *v*/*v* concentration, while the DC-Cu-Gly and LDH-Fe-Gly effluents showed 100% mortality up to 50% *v*/*v* concentration (Figure 3B). The LDH and DC effluents impregnated with Fe and Cu showed toxicity levels up to 10% to1% (*v*/*v*) concentrations. These results are similar to those reported by Cuhra [61], who evaluated a commercial formulation of glyphosate (*Roundup*) and where acute toxicity levels are observed even at concentrations below 10 mg/L that induce mortality of *D. magna* within 48 h [61]. Consistent with previous studies, commercial glyphosate formulations such as the one used in this study may contain adjuvants and/or additives at various concentrations that may increase toxicity compared to the active ingredient glyphosate [61,62]. *D. magna*, compared to the glyphosate and amoxicillin treatments, presented statistical differences only in samples after 25% (*v*/*v*).

Similarly, some authors report variable rates of glyphosate absorption in other invertebrates, such as *D. pulex*, suggesting small and short-term effects in the food chain, given the prevalence of glyphosate in the aquatic environment [63]. These studies illustrate the importance of using test organisms in ecotoxicology studies for more accurate interpretation of toxicity data.

In electro-Fenton processes, the removal of toxicity in mining effluents has been achieved, with *D. magna* [64] as the sole evaluated indicator. Furthermore, a reduction in toxicity has been observed in wastewater treated through photo-Fenton and UV/H_2_O_2_/RWW in some organisms, although contrasting results have been obtained with other indicators [65]. On the other hand, in water samples containing microplastics treated with photo-Fenton, neither toxicity nor lethality has been observed due to the absence of the development of toxic by-products during the degradation process [66]. Regarding studies of photocatalysis with TiO_2_ in water, effects have been reported in other animal models, such as *Hydra attenuata*, where lethality rates of 13% were recorded at concentrations of 6.25% (*v*/*v*) [42]; however, it is important to mention that lethality prior to treatment ranged from 100% to 50% (*v*/*v*).

### 3.5. Mutagenicity Index (MI) Using the Ames Test

Figure 4 shows the results of the mutagenicity test of the treated and untreated effluents with TA98 and TA100 strains. Figure 4A shows the results obtained from the treated and untreated effluents containing amoxicillin. It is evident that most of the samples did not show mutagenicity, except for the effluent samples treated with LDH-Fe, as in the case of the TA98 strain.

In Figure 4B, it was observed that the samples catalyzed with double-layer hydroxides presented mutagenicity, with the difference that when the sample was diluted by 50%, the effluent catalyzed with LDH-Fe ceased to be mutagenic; however, this did not occur with the samples catalyzed with LDH-Cu, since this showed mutagenicity even when they were at a concentration of 1% (*v*/*v*). In Figure 4C,D, neither the starting compound nor the treated samples show mutagenicity to either strain (TA98 or TA100). Significant differences were observed between *Salmonella* strains TA98 and TA100 in 100% (*v*/*v*) glyphosate matrices. However, strain TA98 and TA100 in amoxicillin did not show statistical differences.

It can be seen that amoxicillin and the effluents treated with LDH showed mutagenicity; those that were presented were the treatments in which the laminar structure was impregnated with iron and copper metals, suggesting that even in the low concentrations in which these were found in solution and whose values are shown in Table 1, may be the main cause of genetic damage [67]. In addition, it was demonstrated that the samples treated with LDH-Fe showed a mutagenic effect in the two strains (TA98 and TA100), indicating that two types of mutations were expressed, one-part frameshift and the other frameshift substitution bases [68]. In contrast, samples treated with LDH-Cu showed mutagenic potential only for strain TA100, indicating only frameshift-type mutations [69]. In addition, it was shown that the LDH-Cu catalyzed effluent was mutagenic for the TA100 strain but not for the TA98 strain, as the TA100 strain has been shown to be the most sensitive in these studies [70].

Aromatic amines have been found to be by-products of oxidation processes that degrade amoxicillin. This was reported by Xie et al. [71], who used a heterogeneous Fenton process to degrade amoxicillin by incorporating Fe° into organometallic compounds (MOFs). Aromatic amines were among the reported products [68]. In addition to the by-products revealed by Xie [68], the study carried out by Pan and Sun [72], in which they evaluated a graphite-catalyzed electro-Fenton compound doped with Cu° and Cu^+^ for the removal of amoxicillin, showed the presence of aromatic amines among the products of the reaction structures [72], which could possibly be responsible for the mutagenicity, since in the report by Handi [73] these functional groups are mutagenic for both the TA98 strain and for the TA100 [73].

It was observed that neither glyphosate nor its degradation by-products generated by the Fenton-type oxidation process exhibited any mutagenic effect, as indicated by the absence of index values equal to or greater than two in any dilution prepared. This finding confirms that the pesticide is in compliance with the regulations that state that phytosanitary products must not induce genotoxicity since a positive mutagenicity test would be a reason to prohibit the use of different commercial formulations [74].

The treatment of textile effluents, after being subjected to gamma radiation and hydrogen peroxide, achieved a 59% reduction in mutagenicity for effluents treated with gamma radiation and a 54% reduction in the case of strains TA98 and TA100, respectively, compared to the mutagenicity observed before treatment [75]. In the case of treatments involving gamma radiation and hydrogen peroxide for non-ionic surfactant compounds, a decrease in mutagenicity has been observed, as reflected in the indices recorded by strains TA98 and TA100 [76]. According to Stalter et al. 2010 [77], it has been reported that the genotoxicity of wastewater effluents increases after ozone treatment. Finally, strategies have been proposed to eliminate the toxicity of wastewater after advanced oxidation processes (AOPs). It is crucial to pay attention to changes in the toxicity of wastewater to assess the performance of AOPs [78].

## 4. Conclusions

Fenton-type processes catalyzed with delaminated clays and double-layer hydroxides impregnated with Fe or Cu are effective in removing amoxicillin and glyphosate. However, the removal process produces effluents that are toxic to various organisms. In addition, not all organisms react in the same way to different catalysts, so the same catalyst may have different effects on *L. sativa*, *D. magna*, and *S. capricornutum*. In most bioassays, toxicity decreased when the effluent was diluted to a 50% (*v*/*v*) concentration. The mutagenicity tests showed that the treatments and the glyphosate compound were not mutagenic. However, when the treated molecule was amoxicillin catalyzed with double-layer hydroxides with Fe and Cu, mutagenicity was found, highlighting the need for careful consideration of this process. To ensure environmental safety, the effluents generated by the treatment process should be subjected to subsequent treatment before discharge into surface waters, as they have been found to exhibit phytotoxicity in *L. sativa*, inhibition of *S. capricornutum*, lethality in *D. magna* and, in some cases, mutagenicity.

## Figures and Tables

**Figure 1 ijerph-20-07172-f001:**
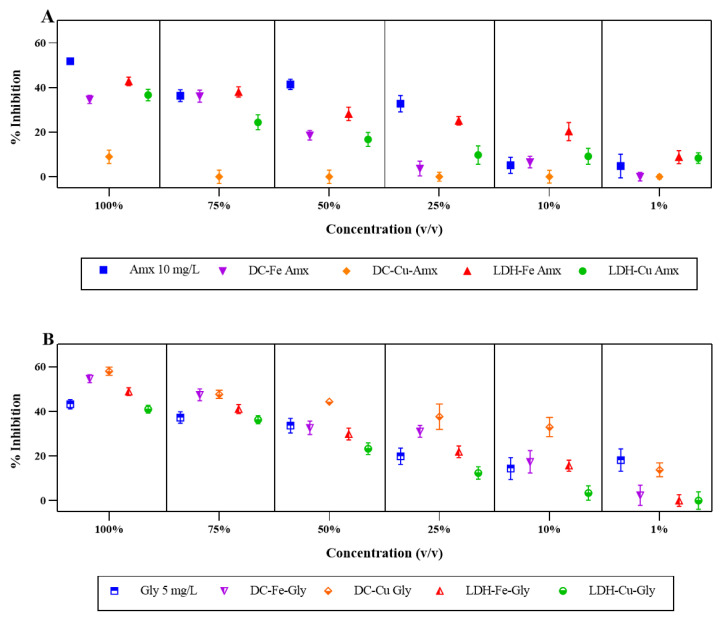
Inhibition percentages of *L. sativa* presented by the treated and untreated solutions of contaminants. (**A**) Percentage inhibition of *L. sativa* with the contaminant amoxicillin (Amx). (**B**) Percentage inhibition of *L. sativa* with the contaminant glyphosate. The removal percentages obtained by a Fenton-type process for amoxicillin were DC-Fe (89%), DC-Cu (80%), LDH-Fe (100%), and LDH-Cu (100%), and for glyphosate, they were DC-Fe (26%), DC-Cu (88%), LDH-Fe (30%), and LDH-Cu (71%).

**Figure 2 ijerph-20-07172-f002:**
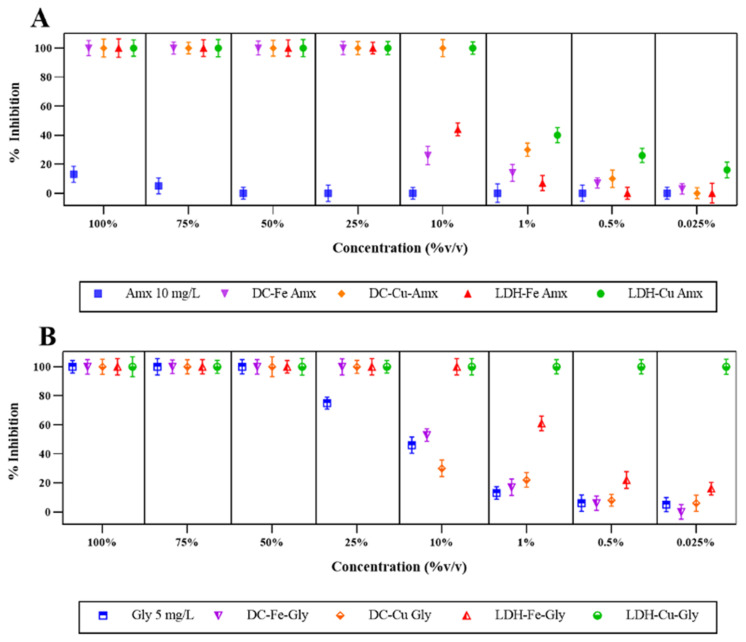
Inhibition percentages of *S. capricornutum* presented by the treated and untreated solutions of contaminants. (**A**) Percentage inhibition of *S. capricornutum* with the contaminant amoxicillin (Amx). (**B**) Percentage inhibition of *S. capricornutum* with the contaminant glyphosate (Gly). The removal percentages obtained by a Fenton-type process for amoxicillin were DC-Fe (89%), DC-Cu (80%), LDH-Fe (100%), and LDH-Cu (100%), and for glyphosate, they were DC-Fe (26%), DC-Cu (88%), LDH-Fe (30%), and LDH-Cu (71%).

**Figure 3 ijerph-20-07172-f003:**
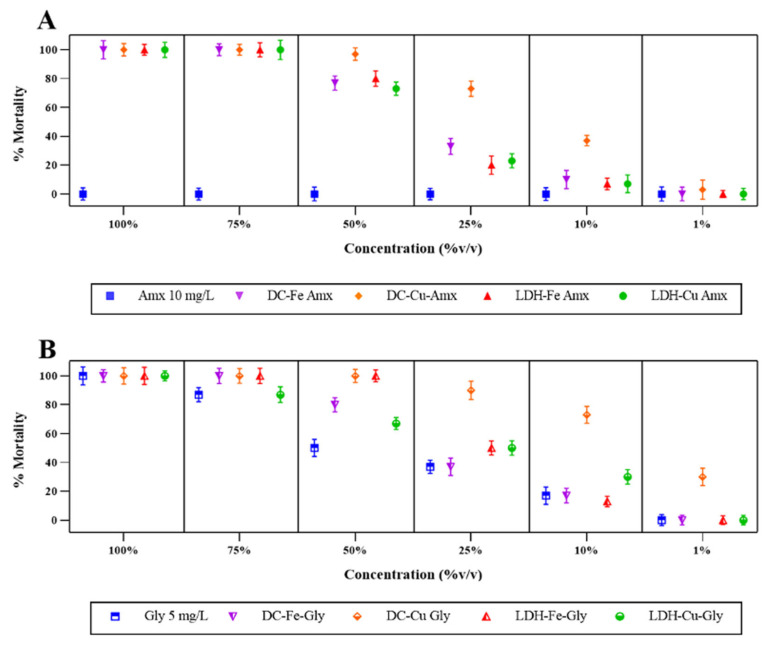
Mortality percentages of *D. magna* presented by the treated and untreated solutions of contaminants. (**A**) *D. magna* mortality rates when the contaminant was amoxicillin (Amx). (**B**) *D. magna* mortality rates when the contaminant was glyphosate (Gly). The removal percentages obtained by a Fenton-type process for amoxicillin were DC-Fe (89%), DC-Cu (80%), LDH-Fe (100%), and LDH-Cu (100%), and for glyphosate, they were DC-Fe (26%), DC-Cu (88%), LDH-Fe (30%), and LDH-Cu (71%).

**Figure 4 ijerph-20-07172-f004:**
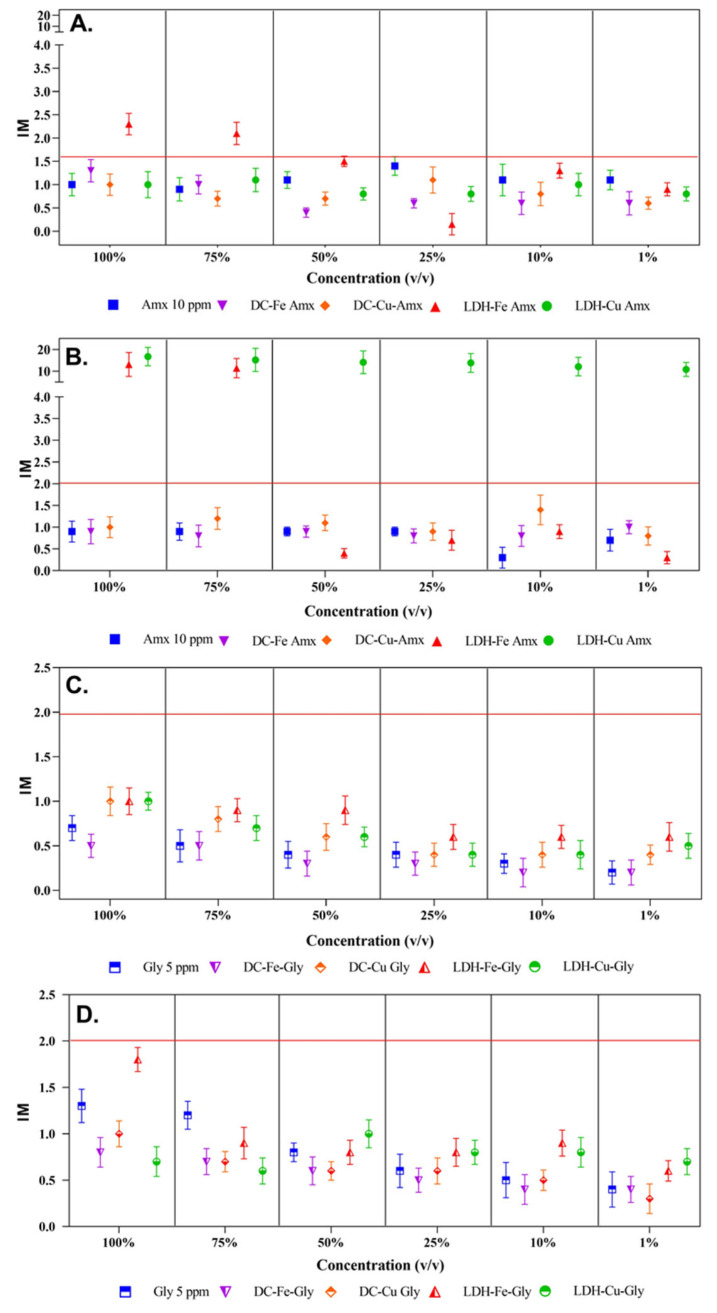
Mutagenicity index (MI) in *S. typhimurium* (TA98 and TA100) presented by the treated and untreated solutions of contaminants. (**A**) Mutagenicity indices of *S. typhimurium* TA98 and TA100, respectively, when the contaminant was amoxicillin (Amx). (**B**) *S. typhimurium* mutagenicity indices TA98 and TA100, respectively, when the contaminant was glyphosate (Gly). The removal percentages obtained by a Fenton-type process for amoxicillin were DC-Fe (89%), DC-Cu (80%), LDH-Fe (100%), and LDH-Cu (100%), and for glyphosate, they were DC-Fe (26%), DC-Cu (88%), LDH-Fe (30%), and LDH-Cu (71%). (**C**,**D**) Neither the starting compound nor the treated samples show mutagenicity to either strain (TA98 or TA100). Significant differences were observed between *Salmonella* strains TA98 and TA100 in 100% (*v*/*v*) glyphosate matrices. However, strain TA98 and TA100 in amoxicillin did not show statistical differences.

**Table 1 ijerph-20-07172-t001:** Physicochemical parameters and percentages of inhibition, lethality, and mutagenicity index of the effluents before and after the removal of amoxicillin and glyphosate using lamellar structures as catalysts of the Fenton-type process.

Catalyst— Pollutant	% Contaminant Removal	Active Phase Leaching mg/L	Final Reaction pH	% Inhibition *L. sativa*	% Inhibition *S. capricornutum*	% Lethality *D. magna*	Mutagenicity Index	
							TA 98	TA 100
DC-Fe Amx	89	0.9	6.90	34.69	100	100	<2	<2
DC-Cu Amx	80	8.2	7.65	8.94	100	100	<2	<2
LDH-Fe Amx	100	0.06	8.48	42.76	100	100	>2	>2
LDH-Cu Amx	100	0.6	8.56	36.65	100	100	<2	>2
AMX 10 mg/L	-	-	-	51.86	13	0	<2	<2
DC-Fe Gly	26	0.2	7.22	54.68	100	100	<2	<2
DC-Cu Gly	88	3.9	7.23	58.04	100	100	<2	<2
LDH-Fe Gly	30	0.6	6.75	48.89	100	100	<2	<2
LDH-Cu Gly	71	0.3	6.98	40	100	100	<2	<2
GLY 5 mg/L	-	-	-	43.23	100	100	<2	<2

## Data Availability

Data are contained within the article and Appendix A.

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
