# Peer review of "Ecotoxicology Evaluation of a Fenton—Type Process Catalyzed with Lamellar Structures Impregnated with Fe or Cu for the Removal of Amoxicillin and Glyphosate"

_ijerph, 2023, doi:10.3390/ijerph20247172_

Round 1
Reviewer 1 Report
Comments and Suggestions for Authors
The manuscript investigated the ecotoxicity of by products produced by treated and untreated effluents, and meaningful results were found that both the two treatments generated mutagenicity. These finds guided the suggestions of suitable measures to reduce the ecological risk. However, some minor revisions should be made before considering acceptance.
Detailed comments were listed as followings:
Figure 4 is too small, so that some useful data cannot be seen clearly.
Author Response
Dear reviewer, thank you very much for the evaluation.
In response, graph 4 increased in size.
Best regards
Reviewer 2 Report
Comments and Suggestions for Authors
Manuscript ijerph-2589085 entitled “Evaluation of the ecotoxicity and mutagenicity index of a Fenton-type process catalyzed with lamellar structures impregnated with Fe and Cu for the removal of amoxicillin and glyphosate” is worthy of investigations and within the scope of MDPI International Journal of Environmental Research and Public Health, Section -Environmental Science and Engineering, Special Issue - Water Pollution and Treatment of Emerging Contaminants.
The manuscript by authors Lugo et al. in the present study reports the results of the evaluation of toxicity produced by treating water containing amoxicillin or glyphosate with a Fenton- type process (AOPs) catalyzed with lamellar structures impregnated with Fe or Cu, using bioassays of Lactuca sativa (seed germination), Selenastrum capricornutum (microalgal growth), Daphnia magna (acute toxicity), and Ames test (mutagenicity).
The use of emerging pollutants such as pharmaceuticals and pesticides has increased significantly worldwide in recent years and therefore enjoys special attention. Advanced oxidation processes (AOPs) offer a promising alternative for the removal of various compounds, including emerging contaminants.
Based on the results of bioassays with four different AOPs treatment catalyzed by the Fenton process with Fe and Cuimpregnated double-layered clay and hydroxide lamellar structures, the authors conclude that such treatment is very efficient in removing amoxicillin and glyphosate contaminants. However, the resulting effluents may be toxic to various organisms due to the inherent toxicity of the untreated contaminants and the advanced oxidation process. To ensure environmental safety, the authors suggest that the effluents resulting from such treatment should be subjected to further treatment before discharge to surface waters.
Minor comments:
Page 1, Line 6, 7, 11
- “y” -> “and”, remove double affiliation marks
Line 18
- Remove AOPs abbreviation form Abstract
Line35
- “physico-chemical”
Page 3, Line114
- you have one bracket excess
Line 130
-> 2.3.1 Seed germination – Lactuca sativa
Line 120-121
-Delete double names of species
Line 143
-> 2.3.2 Algal growth – Selenastrum capricornutum
Line 175, 181 etc
- delete-write temperature and rpm without values in brackets “ at 37 (±2)”
Line 186-192
- Delete left template M&M instructions
The Materials and Methods should be described with sufficient details to 186 allow others to replicate and build on the published results. Please note that the publication of your manuscript implicates that you must make all materials, data, computer code, and protocols associated with the publication available to readers. Please disclose at the submission stage any restrictions on the availability of materials or information. New methods and protocols should be described in detail while well-established methods can be briefly described and appropriately cited.
Comments on the Quality of English LanguageCheck whole MS for typing errors and consistency.
Author Response
Dear Reviewer,
Thank you very much for the review.
The requested changes were made and the paragraph suggested for deletion was removed. The changes are highlighted in Green on the document.
Best Regards.

Reviewer 3 Report
Comments and Suggestions for Authors
review
The present study was supposed to aims to evaluate the toxicity generated by treating water containing amoxicillin or glyphosate with a Fenton- type process catalyzed with lamellar structures impregnated with Fe or Cu, using bioas says of Daphnia magna, Lactuca sativa, Selenastrum capricornutum, and mutagenicity. The study is interesting, but requires improvement before being allowed to proceed to further stages of the publication process.
1. The authors described in detail the research methods used, however the obtained test results were not subjected to statistical analysis. The reader does not know whether the differences in the obtained results are statistically significant?. Please supplement your research with statistical analysis. This will facilitate the interpretation of the obtained research results.
2. The authors should standardize the concentration units used, I suggest using the mg∙dm-3 unit,
3. The conclusions need improvement, in my opinion they need to be more precise.
Author Response
Dear Reviewer;
Thank you very much for your review.
- For each organism, Anova analysis was included relating the catalysts and each treatment and establishing if there are significant differences. These analyzes are included as supplementary material and are displayed at the end of the results of each of the organisms evaluated.
- Dear Evaluator, we have changed units to mg/L throughout the text, however, in the report of dilutions we always place them in V/V, considering that these tests, to establish the limit of inhibition or growth, are dilutes the matrix as a whole.
- The ideas were reorganized to give greater clarity and depth to the proposed conclusions.
Best Regards.
Reviewer 4 Report
Comments and Suggestions for Authors
Major revision

Comments on the Quality of English LanguageEnglish language is satisfactory. Howver, the authors should go through the manuscript.
Author Response
Dear Reviewer;
Thank you very much for your review.
- The title was modified without losing the essence of the work
-
Numerical values were included and grammar was reviewed.
- There are very few ecotoxicological studies on Fenton-type treatments for the molecules used in this study. For the discussion, the existing ones were considered. A new search for papers was carried out to verify if there were new ones but there are none.
-
The ideas were reorganized to give greater clarity and depth to the proposed conclusions.
-
This was reviewed
Round 2
Reviewer 3 Report
Comments and Suggestions for Authors
Review
In my opinion, the authors improved the manuscript in line with my suggestions. The manuscript may be admitted to the next stage of processing.
Reviewer 4 Report
Comments and Suggestions for Authors
The manuscript is accepted in current form. However, during the final revision check grammatical and other syntax errors.
Comments on the Quality of English LanguageMinor editing is required during revision.